# CO$_2$ Emissions in G20 Nations through the Three-Sector Model

**Kejia Yan** [1],*[iD], **Rakesh Gupta** [2][iD] **and Victor Wong** [2]

1    School of Management, Xiamen University, Xiamen 361005, China
2    Department of Accounting, Finance and Economics, Griffith University, Nathan 4111, Australia
*    Correspondence: kejia.yan@griffith.edu.au

**Abstract:** This paper examines the relationship between CO$_2$ emissions in three economic sectors of G20 member countries using the environmental IPAT model and STIRPAT model and validates the EKC hypothesis by comparing the results for developing and developed countries. The results confirm that there is a significant long-run equilibrium relationship between the three sectors (primary, secondary, and tertiary) and CO$_2$ emissions across the panel. Furthermore, the long-run elasticities suggest that the primary sector (agriculture) positively and negatively affects the CO$_2$ emissions of developing and developed economies, respectively. This finding proves that the development of agriculture is in line with the EKC hypothesis that a more developed economy will instead improve environmental degradation. Based on the findings, for each sector, we provide policymakers with suggestions to potentially curb CO$_2$ emissions without significantly compromising economic growth.

**Keywords:** three-sector model; EKC; G20 nations; curbing CO$_2$ emissions; economic growth





## 1. Introduction

Increasing concentrations of carbon dioxide (CO$_2$) emissions in the atmosphere have become a serious global concern, with CO$_2$ emissions being the main cause of many diseases and of temperature and sea level increases. Continued emissions will lead to further warming and long-term changes in the world's climate system, increasing the likelihood of severe, widespread and irreversible impacts on ecosystems (UN-Water 2020). Even though the reduction in CO$_2$ emissions is a global challenge, the standard of responsibility varies from country to country due to differences in energy efficiency, productivity, the stage of economic development, technological innovation and historical cumulative emissions. Therefore, although the importance of reducing CO$_2$ emissions is universally recognized, the international community has not reached a consensus on how much responsibility each country should assume to deal with these disasters, and this discrepancy is reflected mainly in the conflicts between developing and developed countries.

Some of the many international meetings that have been held to discuss the issue of emissions include the Kyoto Protocol in 1997, the Copenhagen Accord in 2009, and the Paris Agreement in 2015. The aim of these meetings has always been to set targets for reductions and to establish processes to achieve these targets. The most productive result of these meetings was that the parties to the Paris Agreement agreed to aim to control the increase in global average temperatures to between 1.5 and 2 °C (IPCC 2019) above preindustrial levels. However, lengthy negotiations between developing and developed countries (from 2015 to 2021 and not yet concluded) have prompted some scientists to revise the target for global average temperature rise, stating that it must now be limited to 1.5 °C above preindustrial levels or civilization will be seriously threatened (Lenton et al. 2019).

Making purely political compromises that disregard market realities may mean success for the Paris Agreement, but governments are only willing to actively cut carbon emissions if their economies do not suffer. The issue at stake is that CO$_2$ emissions are associated with economic activities, while a reduction in CO$_2$ emissions is associated with a sacrifice of economic growth. Therefore, reducing CO$_2$ emissions without sacrificing

economic development is the most acceptable solution for policymakers, as politicians cannot afford the political and developmental costs of significantly reducing carbon emissions at the domestic level. The environmental Kuznets curve (EKC) hypothesis provides the most desirable answer for humanity: "The solution to pollution is economic growth" (Roca et al. 2001).

The impact of economic growth on environmental degradation before and after the turning point of the inverted U-shaped curve implied by EKC is explained by academics in terms of scale, composition and technology, consistent with the three-sector model of economics. That is, in the preindustrial and industrial period, the labor force was concentrated in the primary sector (agriculture) and the secondary sector (manufacturing and industry), and economic growth was heavily dependent on the continuous expansion of production. This situation led to the consumption of large amounts of natural resources and rapid environmental degradation. The huge demand for natural resources has spurred sharp price increases and rising production costs. In this context, technology and productivity upgrades were used to improve energy efficiency and reduce energy consumption to stimulate economic growth. Environmental quality began to improve steadily, with increased levels of environmental awareness, regulation and education. As technology continues to advance and the economy becomes more reliant on services, the primary and secondary sectors will become more environmentally friendly (Panayotou 1993; Grossman 1995; Stern 2004).

A number of scholars have critically modeled the EKC hypothesis; the most recent review of the EKC hypothesis was conducted by Shahbaz and Sinha (2019) on 171 EKC studies of $CO_2$ emissions from 1991 to 2017. However, the findings of these studies suggest that the EKC results are doubtful with respect to the choice of control variables, pollution indicators, and environmental and methodological adaptation. Indeed, there are many other factors could influence the EKC results, such as government policy change, cumulative effect on emissions, and some special issues such as the 1980 oil crisis and global financial crisis. For this reason, the EKC hypothesis remains controversial in academic circles.

Significantly, scholars determine whether the EKC is effective by verifying the quadratic relationship[1] between economic and environmental variables (Dogan and Turkekul 2016; Yasin et al. 2020). Although the three effects of size, composition and technology in the context of the three-sector model play a very important role in the explanation of the EKC hypothesis and are generally accepted, the validity of the EKC hypothesis has rarely been tested by validating the relationship between the three-sector model and $CO_2$ emissions in developing and developed countries directly. That is, if economic growth reduces carbon emissions across all three sectors (primary, secondary and tertiary sectors), do developed and developing countries have the opposite result? The primary sector (agriculture), for example, produces more $CO_2$ emissions in developing countries but reduces $CO_2$ emissions in developed countries. In comparison to the many controversial validation formulas to determine whether EKC is valid, the three-sector model of validation can combine the three EKC effects with real industries to identify their patterns.

Moreover, few studies have employed the economic three-sector model in the area of $CO_2$ reduction. Previous studies in this area have focused on the relationship between specific variable(s) from one or two sectors and $CO_2$ emissions in one country or region (Shahbaz et al. 2013; Paramati et al. 2017; Dong et al. 2018). Motivated by these factors, this study aims to fill this research gap by identifying the factors that influence $CO_2$ reduction and economic development for different groups of economies within the context of the three-sector model. Furthermore, this is the first paper to test the EKC hypothesis by comparing the results of the impact of different sectors on $CO_2$ emissions in developed and developing countries. A better understanding of the relationship between these three economic sectors and $CO_2$ emissions can guide countries in reducing $CO_2$ emissions without significantly compromising economic growth and accelerate global cooperation in $CO_2$ reduction and investment (mutual wins with regard to the environment and the economy), further helping negotiations between developing and developed countries on the issue of emissions.

This paper selects variables that account for economic growth, $CO_2$ emissions, energy consumption, and technological advancement. Twelve variables[2] from the three sectors were chosen to test the elasticity between these variables and $CO_2$ emissions in G20 countries to explain the relationships among $CO_2$ emissions and the stage of development. The G20 countries are used as a data sample because their members cover a wide range of areas and are representative of the global economy. G20 members account for approximately 78% of GHG emissions (IPCC 2019) and 90% of the global economy, including 80% of the global trade volume (Burck et al. 2015). Thus, if the G20 members do not act, any impact on the emissions gap is likely to be very limited.

## 2. Literature Review

In the three-sector model, the development structure of a country's economy should evolve from the primary sector to the secondary sector and, finally, to the tertiary sector. The differences between each sector are reflected in workforce participation, technology, capital and development gaps (Bah 2007). These gaps affect how economic resources are reallocated in the economy and the distribution of pollution from country to country. The primary sector tends to make up a larger percentage of economies in developing economies than it does in developed economies. On average, developing economies distribute 6.5 times more of the workforce in agriculture than developed economies do (World Bank 2021). As the primary and secondary sectors become more technologically advanced in developed economies, productivity from resources results in greater yield per unit of resource input. Higher productivity and higher yield result in higher labor productivity and energy efficiency, resulting in more environmentally friendly production.

For developed economies, the productivity gap between sectors is small; economic growth results mainly from higher productivity and higher energy efficiency but not from the structural change from the primary sector to the secondary sector, for example, leading to the obsolescence of less productive firms caused by updating technology (Aghion and Howitt 1992; McMillan and Rodrik 2011; Roncolato and Kucera 2013; Morsy et al. 2015; Gai et al. 2021; Hamory et al. 2021).

Nevertheless, structural change, especially from the primary to the secondary sectors, has played a significant role in the economic growth of developing economies, particularly in the reallocation of labor from low-productivity sectors to higher-productivity sectors in modern service and manufacturing industries (Timmer and Akkus 2008). As agriculture formerly occupied a large share of the workforce, the possibility of labor reallocation through structural change provides opportunities. Structural change contributes to economic growth, a rise in labor productivity, an increase in the employment rate through quality jobs, a decrease in the poverty rate, an increase in per capita income and an increase in energy productivity (McMillan and Rodrik 2011; Morsy et al. 2015; Gai et al. 2021; Hamory et al. 2021).

Thus, the key areas of $CO_2$ emissions differ based on the essential differences in economic growth between developing and developed countries. The distribution of $CO_2$ emissions by industry sector for G20 countries in 2014[3] (World Bank 2021) is exhibited in Table 1. This table indicates that the primary sector and secondary sectors of developing countries (2.64% and 67.41%) generated more $CO_2$ emissions than those in developed countries (1.68% and 56.70%). The tertiary sector contributed 41.61% of $CO_2$ emissions from developed countries, and this number for developing countries was approximately 30%. Thus, different distributions of sector $CO_2$ emissions indicate different directions in the environmental governance of $CO_2$ reduction for developing and developed countries in the G20.

**Table 1.** $CO_2$ emissions by industry sector for G20 countries in 2014.

| G20 Countries | Primary Sector | Secondary Sector | | Tertiary Sector | |
| --- | --- | --- | --- | --- | --- |
| | Other Sectors Including Agriculture (%) | Electricity and Heat Production (%) | Manufacturing Industries and Construction (%) | Transport (%) | Residential Buildings and Commercial and Public Services (%) |
| Developing countries in the G20 | | | | | |
| Argentina | 6.47% | 38.04% | 16.87% | 24.17% | 14.46% |
| Brazil | 4.05% | 26.31% | 20.60% | 44.75% | 4.29% |
| China | 2.07% | 52.25% | 31.72% | 8.60% | 5.36% |
| Indonesia | 3.02% | 53.61% | 26.41% | 11.48% | 5.49% |
| India | 1.43% | 44.25% | 18.40% | 30.81% | 5.11% |
| Mexico | 2.07% | 44.07% | 13.45% | 35.09% | 5.32% |
| Russia | 1.15% | 61.11% | 12.32% | 16.24% | 9.17% |
| Saudi Arabia | 0.00% | 49.16% | 24.10% | 25.92% | 0.82% |
| Turkey | 3.71% | 46.69% | 14.62% | 19.83% | 15.16% |
| South Africa | 2.42% | 67.48% | 12.58% | 12.05% | 5.47% |
| Average | 2.64% | 48.30% | 19.11% | 22.89% | 7.07% |
| Developed countries in the G20 | | | | | |
| Australia | 1.69% | 58.36% | 11.49% | 24.74% | 3.72% |
| Canada | 2.91% | 38.73% | 12.04% | 31.79% | 14.52% |
| Germany | 0.05% | 48.47% | 12.44% | 21.37% | 17.67% |
| France | 4.70% | 13.80% | 15.70% | 42.41% | 23.40% |
| UK | 0.96% | 41.93% | 9.60% | 28.45% | 19.06% |
| Italy | 2.23% | 35.56% | 11.19% | 32.95% | 18.07% |
| Japan | 0.21% | 53.10% | 19.18% | 17.54% | 9.98% |
| South Korea | 1.45% | 60.49% | 13.66% | 16.28% | 8.13% |
| USA | 0.94% | 45.99% | 8.66% | 33.40% | 11.01% |
| Average | 1.68% | 44.05% | 12.66% | 27.66% | 13.95% |

### 2.1. Primary Sector and $CO_2$ Emissions

In most of the developing countries, the highest proportion of the workforce and the generation of a large proportion of income are distributed throughout the production of agriculture, raw materials, livestock, fishing and forestry, which together make up the primary sector. Agriculture is an industry that uses the growth and development of plants and animals to obtain products through artificial cultivation; it is one of the basic industries that supports the construction and development of a national economy. For example, in Nigeria (Odetola and Etumnu 2013; Oluwole et al. 2021), India (Mahadevan 2003; Orhan et al. 2021), Indonesia (Adebayo et al. 2021), China (Yao 2010; Li et al. 2021), Northern Cyprus (Katircioglu 2006), and developing countries (Awokuse 2009; Naseem and Tong 2021), agriculture has contributed positively and consistently to economic growth.

The technology gap in agricultural practices among developed and developing countries has widened, especially in output per hectare and per worker. When the primary sector becomes more technologically advanced in developed countries, the productivity levels and yields for the same crop will be much higher than in developing countries (Ruttan 2002; Mohsin et al. 2021). Ball et al. (2016) confirmed that the main contributor to economic growth in the US agriculture sector is productivity growth. Labor productivity can be improved by improvement in technologies and can create labor, providing workers for growing urban areas (Huffman and Orazem 2007). Furthermore, advanced technology will help maintain a rational energy structure in developed economies. Possible technologies that can create savings in energy of 10% to 40% include reduced soil tillage, optimized fertilizer efficiency, improved irrigation techniques and enhanced solar drying. Instead, fossil fuels with biomass will slow down $CO_2$ emissions; between 0.25 and 1 Gt of fossil fuel carbon could be replaced by agricultural biofuels per year (Sauerbeck 2001).

However, compared to $CO_2$ reduction, developing countries are more committed in poverty reduction and in the development of agriculture. In their analysis of multiyear data for 29 developing countries, Webb and Block (2012) indicated that when agriculture is supported by governments, poverty levels fall faster.

Consequently, agriculture may increase $CO_2$ emissions in developing countries and decrease $CO_2$ emissions in developed countries. Previous research has focused on the relationship between agriculture, the economy and technology for a specific country. As such, the relationship between the primary sector and $CO_2$ emissions has not been examined in a comprehensive manner.

*2.2. Secondary Sector and $CO_2$ Emissions*

The secondary sector contains middle per capita income countries that generate the main part of their income through industrial production and construction, such as mining, manufacturing, construction, and electricity, heat, energy, and water production and supply. The importance of the manufacturing industry for economies is self-evident; the higher the structure of a country's manufacturing sector is, the better it is able to develop a competitive service economy (Guerrieri and Meliciani 2005; Wu et al. 2021). However, industry impacts $CO_2$ emissions heavily for both developing and developed countries (Tunç et al. 2007; Parikh et al. 2009; Ali 2015; Muangthai et al. 2016; Yuan et al. 2017; Abbasi et al. 2021a). The difference is that developing economies have a more labor-intensive manufacturing goods industry structure than developed countries, while developed economies have a more technology-intensive industry structure (Shafaeddin 2004; Liu et al. 2021; Shen et al. 2022).

These differences impact countries' sensitivity to $CO_2$ emissions. For example, energy is an essential element of economic activity. At the same time, energy is the leading source of greenhouse gases. The global annual $CO_2$ concentration has increased by more than 48% since the start of industrialization in the mid-18th century, when humans' burning of biomass and fossil fuels to provide energy began on a significant scale (Aziz et al. 2013; Shafiei and Salim 2014; Manuel et al. 2016; Crafts and Mills 2017). Furthermore, approximately 80% of energy sources and 66% of electricity sources in the world use nonrenewable energy from fossil fuels (IPCC 2019). Thus, after attempting to determine the positive relationship between energy consumption and economic growth, researchers suggested that countries enact more vigorous energy policies since reducing energy consumption will adversely affect GDP in the long run (Ghosh 2002; Lee and Chang 2008; Zhang and Cheng 2009; Abbasi et al. 2021b; Khan et al. 2021).

More importantly, inefficient energy production, less sophisticated energy production techniques and the large proportion of conventional energy in total energy consumption are the main weaknesses in developing economies. An example of this phenomenon occurs in China; Guan et al. (2009) indicated that by 2008, China was already the world's largest $CO_2$ emitter; however, a large proportion of these emissions could be reduced if China updated its inefficient coal-powered electricity system and coal for central heating in most of the northern regions during winter. Yang et al. (2018) stated that China's industrial sector uses mostly fossil fuel energy. When power generation switches to oil from thermal coal, $CO_2$ emissions decrease and GDP increases in both the short and long term (Zhao et al. 2018).

Intuitively, an increase in the proportion of renewable energy among all energy sources should decrease $CO_2$ emissions. Some scholars have also argued that there is a negative relationship between $CO_2$ emissions and renewable energy consumption (Lee 2013; Romano and Scandurra 2013; Ben Jebli et al. 2014; Shafiei and Salim 2014; Agbelie 2016; Asongu et al. 2016; Manuel et al. 2016; Riti and Shu 2016; Spetan 2016; Dudin et al. 2017; Soylu et al. 2021; Bekun 2022). However, few researchers have indicated that there is a positive relationship between $CO_2$ emissions and renewable energy consumption. Apergis et al. (2010) examined the causal relationship between $CO_2$ emissions and renewable energy consumption for a group of 19 developed and developing countries for the period from 1984 to 2007 using a panel VECM model and found a statistically significant positive relationship between $CO_2$ emissions and renewable energy consumption in the long run. Leitao (2014) investigated

the correlation between $CO_2$ emissions and renewable energy for the period from 1970 to 2010 using time series data on the Portuguese economy and found that $CO_2$ emissions and renewable energy are positively correlated.

The relationship between $CO_2$ emissions and renewable energy consumption should be considered separately for developing economies and developed economies. Table 2 below shows that solid biofuels for traditional uses are the main renewable sources of energy for developing countries such as China (85.11%), Indonesia (82.24%), India (75.85), Saudi Arabia (100%) and South Africa (81.13%). Solid biofuels can be any biological material used as fuel, such as wood, sawdust, leaves, and dried animal dung (World Bank 2021). Thus, the solid biofuels for traditional uses will produce $CO_2$ emissions. Consequently, it is possible that renewable energy has a positive effect on $CO_2$ emissions for developing countries. Conversely, no developed economies use traditional solid biofuels as a source of renewable energy.

**Table 2.** The percentage of different types of renewable energy for all G20 economies in 2012.

| | Country | Solid Biofuels for Traditional Uses | Solid Biofuels for Modern Uses | Hydro Energy | Liquid Biofuels | Wind Energy | Solar Energy | Geothermal Energy | Waste Energy | Biogas Energy | Marine Energy |
|---|---|---|---|---|---|---|---|---|---|---|---|
| Developing Economies | ARG | 7.59% | 37.75% | 47.82% | 6.73% | 0.11% | 0.00% | 0.00% | 0.00% | 0.00% | 0.00% |
| | BRA | 11.40% | 41.23% | 35.28% | 11.70% | 0.09% | 0.27% | 0.00% | 0.00% | 0.03% | 0.00% |
| | CHN | 85.11% | 0.36% | 10.07% | 0.48% | 0.36% | 1.23% | 0.97% | 0.00% | 1.44% | 0.00% |
| | IDN | 82.24% | 15.28% | 1.43% | 0.31% | 0.00% | 0.00% | 0.74% | 0.00% | 0.00% | 0.00% |
| | IND | 75.85% | 20.12% | 3.59% | 0.07% | 0.26% | 0.07% | 0.00% | 0.01% | 0.03% | 0.00% |
| | MEX | 0.00% | 76.75% | 18.37% | 0.00% | 0.23% | 0.67% | 3.96% | 0.00% | 0.02% | 0.00% |
| | RUS | 16.69% | 15.06% | 68.16% | 0.00% | 0.00% | 0.00% | 0.10% | 0.00% | 0.00% | 0.00% |
| | SAU | 100.00% | 0.00% | 0.00% | 0.00% | 0.00% | 0.00% | 0.00% | 0.00% | 0.00% | 0.00% |
| | TUR | 0.00% | 61.69% | 26.00% | 0.22% | 0.80% | 3.06% | 8.07% | 0.00% | 0.15% | 0.00% |
| | ZAF | 81.13% | 17.39% | 1.03% | 0.00% | 0.05% | 0.40% | 0.00% | 0.00% | 0.00% | 0.00% |
| | Average | 46.00% | 28.56% | 21.18% | 1.95% | 0.19% | 0.57% | 1.38% | 0.00% | 0.17% | 0.00% |
| Developed Economies | AUS | 0.00% | 69.56% | 21.43% | 3.13% | 2.29% | 2.43% | 0.00% | 0.00% | 1.17% | 0.00% |
| | CAN | 0.00% | 26.94% | 70.78% | 1.53% | 0.50% | 0.01% | 0.00% | 0.11% | 0.13% | 0.01% |
| | DEU | 0.00% | 47.18% | 12.47% | 12.13% | 11.76% | 3.99% | 0.45% | 5.42% | 6.60% | 0.00% |
| | FRA | 0.00% | 61.36% | 26.35% | 5.73% | 1.32% | 0.32% | 0.77% | 3.23% | 0.72% | 0.21% |
| | GBR | 0.00% | 30.71% | 15.56% | 22.41% | 11.93% | 1.76% | 0.00% | 4.98% | 12.66% | 0.00% |
| | ITA | 0.00% | 24.40% | 48.16% | 11.64% | 3.04% | 2.14% | 7.99% | 1.18% | 1.44% | 0.00% |
| | JPN | 0.00% | 29.86% | 56.54% | 0.00% | 1.00% | 7.39% | 3.59% | 1.62% | 0.00% | 0.00% |
| | KOR | 0.00% | 26.70% | 29.61% | 12.86% | 1.94% | 4.37% | 1.94% | 16.02% | 6.55% | 0.00% |
| | USA | 0.00% | 49.68% | 27.60% | 12.90% | 2.82% | 1.28% | 2.25% | 1.19% | 2.28% | 0.00% |
| | Average | 0.00% | 40.71% | 34.28% | 9.15% | 4.07% | 2.63% | 1.89% | 3.75% | 3.51% | 0.02% |

The replacement of nonrenewable energy with clean renewable energy is a possible solution for policymakers in developing policies for the management of $CO_2$ emissions without compromising economic growth (Ben Jebli et al. 2014; Riti and Shu 2016). Hydroelectricity is the main clean renewable energy source for Argentina (47.82%), Russia (68.16%), Canada (70.78%), Italy (48.16%) and Japan (56.54%). As a form of clean energy, hydroelectricity is highly used in both developing and developed economies. However, other clean energy resources, such as wind, solar, biogas and marine resources, are rarely used in developing economies. For example, almost zero percent of developing economies use marine energy. It is clear that high technology and huge amounts of capital are needed to develop, run and maintain these types of clean energy. Developing economies focus on infrastructure construction and attempt to increase their GDP, paying little attention to $CO_2$ emissions. In comparison, developed economies more consistently use all clean energy resources available. Wind energy accounts for 11.76% and 11.93% of the total renewable energy used in Germany and the United Kingdom, respectively. In Japan, solar energy accounts for 7.39% of the country's total renewable energy usage. Biogas accounts for 12.66% of Germany's renewable energy. Thus, policymakers need to consider $CO_2$ emissions from a global perspective, share their experiences in clean energy use with all countries, and cooperate with each other to break down the technical barriers to clean energy. The innovation and application of technology require significant capital investment; thus, a global green finance platform could also be considered for rapid establishment.

### 2.3. Tertiary Sector and $CO_2$ Emissions

Data for the tertiary sector show that highly developed economies attain a large percentage of their total economic outputs through services, education, tourism and transport. Sectors such as exports, imports, FDI inflows, FDI outflows, commercial services exports, commercial services imports, stock market capitalization and stock market trading value have been chosen as representative of the tertiary sector. Consider variables, exports, imports, commercial services exports and commercial services imports as examples. All of these variables differ in terms of their impact on $CO_2$ emissions in developing and developed countries.

(1)    Exports and imports

Exports and imports are a very important component of a country's economy. Much of the previous literature shows that both exports and imports benefit economic growth at the country level (Fosu 1990; Ekanayake 1999; Awokuse 2007, 2008; Ugur 2008; Achchuthan 2013). Exports and imports enhance economic growth, and economic growth leads to energy demand and an increase in $CO_2$ emissions (Sánchez-Chóliz and Duarte 2004). After an investigation of 30 countries in the European Union between 1991 and 2012, Pie et al. (2018) found that the more a country imports, the higher the country's $CO_2$ emissions will be.

Some scholars have argued that the effect of exports and imports on $CO_2$ emissions is very different between developed and developing economies. Kondo et al. (1998) concluded that in Japan, the amount of $CO_2$ emissions embodied in imports was higher than that embodied in exports. Yan and Yang (2010) mentioned that imports produced 4.4% to 9.05% of China's annual $CO_2$ emissions, while exports produced 10.03% to 26.54% of its annual $CO_2$ emissions from 1997 to 2007. Davis and Caldeira (2010) found that imports to China and the United States were shown to contain 0.49 kg and 0.77 kg of $CO_2$ per dollar on average, respectively. They also found that in some Western countries, such as Switzerland, Sweden, Austria, the United Kingdom, and France, more than 30% of $CO_2$ emissions came from imports.

It seems that imports produce more $CO_2$ emissions in developed economies and that exports produce more $CO_2$ emissions in developing economies, which can be interpreted to mean that developing economies and developed economies are at different stages of development. The early stages of manufacturing will produce more $CO_2$ emissions than product innovation will. This finding can also be interpreted to mean that developing economies import $CO_2$ to developed countries and developed economies export $CO_2$ to developing countries (Yan and Yang 2010). Some scholars further suggested that developed economies should reduce their domestic emissions by increasing imports from developing economies (Weber and Matthews 2007; Wiedmann et al. 2008; Guan et al. 2009; Baiocchi and Minx 2010). These emissions may be transferred to other countries, such as China, as imports. Li and Hewitt (2008) indicated that 4% of China's $CO_2$ emissions are caused by consumption from the UK. Shui and Harriss (2006) found that approximately 7% to 14% of China's $CO_2$ emissions are due to consumption from the US. Thus, imports may reduce the domestic pollution level in developed economies.

(2)    Commercial services

Typically, commercial services[4] in the tertiary industry include high-tech products and services, such as information technology (IT), communication and telecommunication services. These advanced products and services generate high productivity for various economic sectors (Guerrieri and Meliciani 2005). As early as 1965, Fuchs (1965) discussed the concept of the "services economy", and since the mid-1950s, a low proportion of the U.S. workforce has been in the tangible goods sector. Information and communication technologies have become the basis of economic development strategies in recent years (Gibbs and Tanner 2010). Moreover, information and communication technologies have a significant impact on the trade performance of the manufacturing industry (Guerrieri and Meliciani 2005). Consequently, the growth of an economy is increasingly dependent on the

transmission of complex, unmodifiable information and communication services (Leamer and Storper 2001).

　　There are three reasons why the relationship between commercial services and $CO_2$ emissions differs between developing and developed countries. First, many of these services and products rely on high-tech equipment. Equipment design in developed countries and manufacturing in developing countries are part of a mature supply chain in the modern world. As shown by Yan and Yang (2010), developed countries import finished goods and services; therefore, they export $CO_2$ to developing countries. Second, electricity is the basic driving energy for many commercial services, but the energy structure is very different in developing and developed countries. The largest source of renewable energy in developing countries within the G20 is traditional, non-clean, solid biofuels, and in developed countries, these fuels are hardly used at all. Wee and Choi (2010) indicated that $CO_2$ emissions would be reduced by 87% and 97%, respectively, if solar cell and wind turbine systems, as renewable energy sources, were substituted for traditional electricity sources. Finally, innovation in commercial services happens very quickly. When a developed country invents a new product and updates many generations of that product, the developing country is completely left behind in terms of copyrights and options. More importantly, it is very difficult to offset the technology gap using either effort or capital (Guerrieri and Meliciani 2005). As such, importing and a backward energy structure will create a vicious cycle of $CO_2$ emissions in developing countries.

　　Table 3 below shows the volume of exports, imports, and exports minus imports of commercial services for the G20 economies in 2018[5]. Commercial services exports from developed economies are nine times greater than those from developing economies. Similarly, developed economies import five times more commercial services than developing economies. Developed economies show a USD 740.22 billion surplus of commercial services, while developing economies show a USD 390.03 billion deficit. In 2018, the technology gap between developing and developed economies was more than USD 1000 billion. Table 4[6] below shows the preliminary analysis for the growth rate of commercial service exports for G20 countries from 1997 to 2018. The results indicate that the mean values for all countries are positive, which means that in recent years, commercial service exports have been increasing for all G20 countries. The BRICS countries show the highest mean growth rate among the G20 members: Brazil (11.05%), Russia (8.28%), India (17.7%), China (17.14%), and South Africa (6.72%). Moreover, developed economies show a smoother growth rate, while developing economies show more volatile values. The highest difference between the maximum and minimum values can be up to 223.2% for China. This is shown by skewness and kurtosis: most of the developed economies (90%) show a left skew (skewness is lower than 0), but developing economies show a half left skew and half right skew (skewness is greater than 0). Similarly, the kurtoses for developed economies are close to normal values (kurtosis is 3), but half of the developing economies show leptokurtosis, where kurtosis is greater than 3. This result is also supported by the Jarque–Bera (JB) test results, for which the null hypothesis for the JB test is normally distributed. Since all developed economies have not rejected the null hypothesis above a probability of 0.1%, the result suggests that commercial service export growth rates for all developed economies are normally distributed. In contrast, half of the developing economies rejected the null hypothesis of the JB test under a probability of 0.1%, and the results suggest that these countries are not normally distributed. Results with leptokurtic distributions are conventionally considered to be inherently more flexible than results with normal distributions (Stacey 2008). This evidence shows that commercial services in developing countries may have different reactions to $CO_2$ emissions than those in developed countries. To our knowledge, this is the first study to provide a comprehensive analysis of exports and imports of $CO_2$ emissions by commercial services in G20 nations.

**Table 3.** The volume of exports, imports, and exports minus imports of commercial services for the G20 nations in 2018.

| | Country | Commercial Services Exports | | Commercial Services Imports | | Exports—Imports |
|---|---|---|---|---|---|---|
| | | Billion USD | % G20 Total | Billion USD | % G20 Total | Billion USD |
| Developing Economies | ARG | 13.91 | 0.26% | 23.61 | 0.47% | −9.70 |
| | BRA | 33.22 | 0.62% | 65.73 | 7.56% | −32.50 |
| | CHN | 231.81 | 4.33% | 521.34 | 10.43% | −289.53 |
| | IDN | 27.21 | 0.51% | 34.98 | 0.70% | −7.77 |
| | MEX | 28.81 | 0.54% | 37.51 | 0.75% | −8.70 |
| | RUS | 63.74 | 1.19% | 93.39 | 1.87% | −29.65 |
| | SAU | 17.39 | 0.32% | 55.48 | 1.11% | −38.09 |
| | TUR | 48.19 | 0.90% | 21.77 | 0.44% | 26.43 |
| | ZAF | 15.59 | 0.29% | 16.11 | 0.32% | −0.52 |
| | Total | 479.88 | 8.97% | 869.90 | 17.40% | −390.03 |
| Developed Economies | AUS | 68.64 | 1.28% | 71.61 | 1.43% | −2.98 |
| | CAN | 91.76 | 1.72% | 111.83 | 2.24% | −20.07 |
| | DEU | 337.15 | 6.30% | 363.88 | 7.28% | −26.73 |
| | EUU | 2490.17 | 46.55% | 2109.82 | 42.20% | 380.35 |
| | FRA | 291.92 | 5.46% | 257.36 | 5.15% | 34.56 |
| | GBR | 374.54 | 7.00% | 229.01 | 4.58% | 145.53 |
| | ITA | 120.73 | 2.26% | 123.84 | 2.48% | −3.11 |
| | JPN | 188.94 | 3.53% | 198.91 | 3.98% | −9.96 |
| | KOR | 97.96 | 1.83% | 127.30 | 2.55% | −29.34 |
| | USA | 808.22 | 15.11% | 536.24 | 10.73% | 271.98 |
| | Total | 4870.02 | 91.03% | 4129.80 | 82.60% | 740.23 |

**Table 4.** Summary statistics of the commercial service export growth rate for the G20 countries from 1996 to 2018.

| | Country | Mean | Median | Maximum | Minimum | S.D. | Skewness | Kurtosis | Jarque–Bera | Probability |
|---|---|---|---|---|---|---|---|---|---|---|
| Developing Economies | ARG | 6.89% | 4.51% | 28.38% | −24.33% | 0.1366 | −0.1436 | 2.5455 | 0.2529 | 0.8812 |
| | BRA | 11.05% | 11.05% | 33.20% | −15.69% | 0.1546 | −0.2009 | 1.8155 | 1.3688 | 0.5044 |
| | CHN | 17.14% | 5.98% | 179.50% | −43.70% | 0.4329 | 2.6201 | 10.7609 | 76.7306 | 0.0000 |
| | IDN | 10.75% | 5.11% | 141.85% | −36.10% | 0.3557 | 2.3479 | 10.0299 | 62.5358 | 0.0000 |
| | IND | 17.70% | 18.32% | 59.73% | −12.48% | 0.1569 | 0.5588 | 3.8596 | 1.7394 | 0.4191 |
| | MEX | 3.49% | 3.52% | 17.68% | −41.18% | 0.1323 | −1.9105 | 7.3224 | 29.1221 | 0.0000 |
| | RUS | 8.28% | 10.56% | 29.77% | −21.61% | 0.1543 | −0.5051 | 2.1363 | 1.5458 | 0.4617 |
| | SAU | 11.69% | 7.39% | 90.89% | −43.46% | 0.2503 | 1.2620 | 6.9632 | 19.3181 | 0.0001 |
| | TUR | 7.66% | 10.71% | 49.03% | −30.17% | 0.1913 | −0.1011 | 2.7183 | 0.1052 | 0.9488 |
| | ZAF | 6.72% | 1.49% | 70.00% | −10.87% | 0.1707 | 2.5106 | 10.1130 | 66.3318 | 0.0000 |
| Developed Economies | AUS | 6.40% | 8.05% | 22.40% | −10.85% | 0.0934 | −0.2298 | 2.4443 | 0.4551 | 0.7965 |
| | CAN | 5.84% | 7.22% | 17.66% | −8.75% | 0.0694 | −0.6688 | 2.8953 | 1.5750 | 0.4550 |
| | DEU | 7.75% | 8.17% | 27.61% | −7.46% | 0.0915 | 0.2163 | 2.5845 | 0.3148 | 0.8544 |
| | EUU | 6.97% | 7.76% | 21.65% | −12.74% | 0.0831 | −0.2949 | 3.1482 | 0.3235 | 0.8506 |
| | FRA | 5.84% | 7.32% | 18.99% | −13.58% | 0.0910 | −0.5149 | 2.4961 | 1.1500 | 0.5627 |
| | GBR | 7.22% | 7.52% | 23.36% | −13.19% | 0.0909 | −0.1854 | 2.7048 | 0.1965 | 0.9064 |
| | ITA | 3.33% | 2.64% | 17.88% | −16.31% | 0.0879 | −0.3976 | 3.1614 | 0.5761 | 0.7497 |
| | JPN | 5.33% | 4.72% | 24.93% | −14.58% | 0.0995 | −0.0006 | 2.5103 | 0.2099 | 0.9004 |
| | KOR | 7.15% | 9.07% | 28.72% | −20.49% | 0.1295 | −0.2203 | 2.4987 | 0.3897 | 0.8229 |
| | USA | 6.39% | 6.88% | 17.60% | −5.37% | 0.0578 | −0.1411 | 2.8256 | 0.0964 | 0.9530 |

In conclusion, the composition effect of the EKC hypothesis requires pollution increases as the economic structure shifts from the primary sector to the secondary sector, and pollution decreases as the economic structure shifts towards the tertiary sector (Fan et al. 2019). Different countries are at different stages of development, and they rely differently on economic output, which means the same sector may have a different reaction to $CO_2$ emissions control among different economies. There are possibilities for international complementarity and cooperation in experience and technology for controlling $CO_2$

emissions. Motivated by the three-sector model in economics, this thesis examines the effect of primary, secondary, and tertiary sectors on $CO_2$ emissions across the panels of developing and developed economies of the G20 countries. There have been some studies that employed the economic three-sector model in the area of $CO_2$ emissions reduction. This study extends the existing literature by investigating the relationship between commercial service exports, commercial service imports, FDI outflows, and stock trade total value with $CO_2$ emissions.

## 3. Data and Methodology

### 3.1. Nature of the Data and Measurements

This study uses G20 countries as a data sample to systematically test the relationship between economic growth and $CO_2$ emissions through 12 selected variables for different developmental stages. G20 countries include developed and developing countries in different stages of development, and therefore, the findings can be generalized. Annual data from 1990 to 2014 were obtained from the World Bank Development Indicators[7] online database for G20 countries, including Argentina (AGE), Australia (AUS), Brazil (BAR), Canada (CAN), China (CHN), Germany (DEU), France (FRA), the United Kingdom (GBR), Indonesia (IDN), India (IND), Italy (ITA), Japan (JPN), Korea (KOR), Mexico (MEX), Russia (RUS), Saudi Arabia (SAU), Turkey (TUR), the United States (USA), and South Africa (ZAF). Balanced panel data are matched by years and countries with 500 observations for all G20 countries.

The measurements of the variables are as follows: $CO_2$ emissions (CDE) are measured in per capita kilotons; total population (POP) is in millions; GDP per capita (GDP) is in current US dollars; nonrenewable energy consumption (NREC) is the sum of gas, oil and coal measured in thousands of terajoules; renewable energy consumption (REC) is the sum of hydro, solid biofuels, wind, solar, liquid biofuels, biogas, geothermal, marine and waste measured in thousands of terajoules; exports of goods and services (EXT) are measured as a portion of GDP in current US dollars; imports of goods and services (IMT) are measured as a portion of GDP in current US dollars; commercial service exports (COE) are total service exports minus exports of government services not included elsewhere as a portion of GDP in current US dollars; commercial service imports (COI) are total service imports minus imports of government services not included elsewhere as a portion of GDP in current US dollars; foreign direct investment net inflows (FDINI) are measured as a portion of GDP in current US dollars; and foreign direct investment net outflows (FDINO) are measured as a portion of GDP in current US dollars. Agriculture (AGR) includes forestry, hunting, and fishing, as well as the cultivation of crops and livestock production and is measured as a portion of GDP in current US dollars; industry (IND) includes value added in mining, manufacturing (also reported as a separate subgroup), construction, electricity, water, and gas as a portion of GDP in current US dollars. Stock market capitalization (SMC) is the share price times the number of shares outstanding (including their several classes) for listed domestic companies as a portion of GDP in current US dollars; Stock trade total value (STV) is the total number of shares traded, both domestic and foreign, multiplied by their respective matching prices as a portion of GDP in current US dollars.

It is implied that the variables of this study were measured in different units to avoid the problem associated with distributional properties of the data series and generating elasticities in the regression models; therefore, it was necessary to transform all the variables into natural logarithms (Paramati et al. 2017). Thus, we transformed all the variables into natural logarithms before commencing the empirical analysis[8].

### 3.2. Model Specification

The purpose of this study was to provide policymakers with potential suggestions for reducing $CO_2$ emissions without significantly compromising economic growth. This paper investigates the relationship between $CO_2$ emissions and 12 variables through the three-sector model of economies. To achieve these objectives, the existing environmental



IPAT model ([Ehrlich and Holdren 1971](#); [Raskin 1995](#); [York et al. 2003](#); [Paramati et al. 2016](#), [2017](#); [Ozcan and Ulucak 2021](#)) has been explored to determine the relationship between $CO_2$ emissions and 12 variables for the G20 countries. This model discusses the influences of population, GDP and technology factors on $CO_2$ emissions.

$$I = P \times A \times T \tag{1}$$

where $I$ is the environmental impact sourced from the total population ($P$) of the underlying nation, ($A$) is the economic influence or per capita consumption, and ($T$) is the level of technology efficiency per capita or per dollar of GDP. This model has been further extended by [Paramati et al. (2017)](#) with a stochastic impacts by regression on population, affluence and technology (STIRPAT) model to identify the relationship between renewable energy, stock markets and $CO_2$ emissions. Thus, the following equation for our empirical studies has been built:

$$CDE_{it} = f(GDP_{it}, POP_{it}, Variable_{it}, \varepsilon_{it}) \tag{2}$$

where $CO_2$ emissions ($CDE$) are the environmental impact sourced from the economic influence of GDP, total population ($POP$) and economic variables from the three-sector model and 12 variables ($Variable_{it}$) through the three-sector model of economies[9]. Omitted-variable bias is another issue for EKC hypothesis analysis ([Dogan and Turkekul 2016](#)), and there have been a wide array of control variables disaggregated from total GDP into sectors used in the estimation of EKC. However, none of the studies of any of the sectors has produced a valid, unified EKC result. It is worth noting that there is no consensus in the academic community on the use of combinations of explanatory variables, which may lead to different results for different combinations of the same industry in the same country. For example, [Abdallah et al. (2013)](#) refuted the EKC hypothesis for Tunisia by examining the relationship between transport values added, road transport related energy consumption, road infrastructure, fuel price and $CO_2$ emissions from the Tunisian transport sector during the period 1980–2010. However, [Talbi (2017)](#) confirmed the EKC hypothesis for Tunisia by investigating the impact of energy consumption of fuel, energy intensity of road transport, economic growth, urbanization and fuel rate on $CO_2$ emissions in Tunisia. This indicates that stacking too many variables can make the results extremely unstable. Thus, our model aims at addressing the impact of a single variable from the three-sector model on $CO_2$ emissions by not accounting for any potential control variables. $\varepsilon_{it}$ represents individual country effects. Countries and time periods are indicated by the subscripts $i$ ($i = 1 \ldots \ldots$, $N$) and $t$ ($t = 1 \ldots \ldots$, $N$), respectively.

## 4. Empirical Results and Discussion

### 4.1. Panel Unit Root Test

A unit root test was undertaken to measure whether the sample variable is stationary. To examine the distributional properties and order of integration of the variables, two generations of tests have been developed: a first-generation [Im et al. (2003)](#) panel unit root test (IPS) assumes cross-sectional independence across all units; a second-generation [Pesaran (2003)](#) cross-section augmented Dickey–Fuller (CADF) panel unit root test assumes heterogeneous panels with cross-sectional dependence across all units ([Barbieri 2006](#)). The null hypothesis of a unit root is nonstationary, as opposed to the alternative hypothesis of a stationary series with no unit root ([Barbieri 2006](#)). The results of the IPS and Pesaran's CADF unit root tests confirm that all of the variables are nonstationary at level I(0) and stationary at their first-order differences I(1). This finding implies that all of the variables have the same order of integration and may have a long-run cointegration relationship among all the variables. Moreover, nonstationarity at level I(0) and stationarity of first-order differences I(1) is the requirement of the Fisher-Johansen cointegration test and fully modified ordinary least squares (FMOLS). Thus, the cointegration test for the applied models will be introduced in the next section.

### 4.2. Cointegration (Long-Run Relationship)

The results of the panel unit root test in Table 5 confirm that all of the variables have the same order of integration and that there may be a long-run cointegration relationship among all of the variables. Thus, the Fisher-type Johansen panel cointegration test has been introduced to explore the long-run equilibrium relationship among the variables in Equation (2). The results of this test are reported in Table 6. The results show that there is a significant long-run relationship among the variables in each equation for all G20 countries and for the subsamples of developing and developed G20 economies.

**Table 5.** Panel unit root tests (full sample).

| At Level | IPS Panel Unit Root Test Assumes Cross-Sectional Independence | | Pesaran's CADF Unit Root Test Assumes Cross-Sectional Dependence | |
| --- | --- | --- | --- | --- |
| | **Statistic** | ***p*-Value** | **Z [t-Bar]** | ***p*-Value** |
| CDE | 2.574 | 0.995 | 3.925 | 1.000 |
| GDP | −0.882 | 0.189 | 1.500 | 0.933 |
| POP | 6.556 | 1.000 | 1.293 | 0.902 |
| EXT | −0.266 | 0.395 | 0.935 | 0.825 |
| IMT | −0.314 | 0.377 | 0.903 | 0.817 |
| REC | 0.673 | 0.749 | 0.298 | 0.617 |
| NREC | 2.909 | 0.998 | 3.600 | 1.000 |
| AGR | −0.089 | 0.465 | 1.592 | 0.944 |
| IND | −0.367 | 0.357 | 0.912 | 0.819 |
| SMC | −0.605 | 0.273 | 3.257 | 0.999 |
| STV | 0.001 | 0.500 | 1.287 | 0.901 |
| COE | −0.902 | 0.184 | 1.444 | 0.926 |
| COI | −1.184 | 0.118 | −0.539 | 0.295 |
| FDINI | −0.937 | 0.175 | 0.735 | 0.769 |
| FDINO | −1.043 | 0.149 | 1.035 | 0.850 |
| At first difference | | | | |
| CDE | −7.018 *** | 0.000 | −2.935 *** | 0.002 |
| GDP | −6.356 *** | 0.000 | −3.029 *** | 0.001 |
| POP | −4.373 *** | 0.000 | −3.438 *** | 0.000 |
| EXT | −7.370 *** | 0.000 | −3.736 *** | 0.000 |
| IMT | −8.433 *** | 0.000 | −4.581 *** | 0.000 |
| REC | −8.517 *** | 0.000 | −5.355 *** | 0.000 |
| NREC | −7.682 *** | 0.000 | −3.251 *** | 0.001 |
| AGR | −8.129 *** | 0.000 | −6.789 *** | 0.000 |
| IND | −8.625 *** | 0.000 | −5.576 *** | 0.000 |
| SMC | −14.403 *** | 0.000 | −3.044 *** | 0.001 |
| STV | −6.665 *** | 0.000 | −2.720 *** | 0.003 |
| COE | −7.008 *** | 0.000 | −3.723 *** | 0.000 |
| COI | −6.237 *** | 0.000 | −3.980 *** | 0.000 |
| FDINI | −8.234 *** | 0.000 | −4.271 *** | 0.000 |
| FDINO | −10.264 *** | 0.000 | −6.800 *** | 0.000 |

Note: (1) The symbols *** reveal that the *t*-test is significant at the thresholds of 1%, 5%, and 10%, respectively, which means that the null hypothesis is rejected; (2) the unit root tests are estimated using constant and trend variables; and (3) all tests are explored using Stata.

**Table 6.** Johansen Fisher panel cointegration test.

| | G20 | | | | Developing | | | | Developed | | | |
|---|---|---|---|---|---|---|---|---|---|---|---|---|
| Hypothesized | Fisher Stat. * | | Fisher Stat. * | | Fisher Stat. * | | Fisher Stat. * | | Fisher Stat. * | | Fisher Stat. * | |
| No. of CE(s) | (from Trace Test) | Prob. | (from Max-Eigen Test) | Prob. | (from Trace Test) | Prob. | (from Max-Eigen Test) | Prob. | (from Trace Test) | Prob. | (from Max-Eigen Test) | Prob. |
| Model 1a : $CDE_{it} = f(GDP_{it}, POP_{it}, AGR, \varepsilon_{it})$ | | | | | | | | | | | | |
| None | 315.10 *** | 0.0000 | 249.20 *** | 0.0000 | 214.90 *** | 0.0000 | 168.90 *** | 0.0000 | 165.00 *** | 0.0000 | 108.70 *** | 0.0000 |
| At most 1 | 128.70 *** | 0.0000 | 83.38 *** | 0.0001 | 87.23 *** | 0.0000 | 56.98 *** | 0.0000 | 87.46 *** | 0.0000 | 60.76 *** | 0.0000 |
| At most 2 | 83.72 *** | 0.0001 | 70.77 *** | 0.0019 | 52.03 *** | 0.0001 | 47.70 *** | 0.0005 | 44.80 *** | 0.0012 | 32.74 ** | 0.0360 |
| At most 3 | 55.98 ** | 0.0480 | 55.98 ** | 0.0480 | 26.49 | 0.1501 | 26.49 | 0.1501 | 46.03 *** | 0.0008 | 46.03 *** | 0.0008 |
| Model 1b : $CDE_{it} = f(GDP_{it}, POP_{it}, IND, \varepsilon_{it})$ | | | | | | | | | | | | |
| None | 346.70 *** | 0.0000 | 240.20 *** | 0.0000 | 217.10 *** | 0.0000 | 156.00 *** | 0.0000 | 173.40 *** | 0.0000 | 138.30 *** | 0.0000 |
| At most 1 | 168.70 *** | 0.0000 | 111.60 *** | 0.0000 | 101.80 *** | 0.0000 | 67.27 *** | 0.0000 | 74.44 *** | 0.0000 | 48.23 *** | 0.0004 |
| At most 2 | 102.40 *** | 0.0000 | 82.18 *** | 0.0001 | 59.56 *** | 0.0000 | 47.96 *** | 0.0004 | 44.22 *** | 0.0014 | 33.63 ** | 0.0288 |
| At most 3 | 68.69 *** | 0.0032 | 68.69 *** | 0.0032 | 39.13 *** | 0.0064 | 39.13 *** | 0.0064 | 44.93 *** | 0.0011 | 44.93 *** | 0.0011 |
| Model 2a : $CDE_{it} = f(GDP_{it}, POP_{it}, REC, \varepsilon_{it})$ | | | | | | | | | | | | |
| None | 355.80 *** | 0.0000 | 272.90 *** | 0.0000 | 244.90 *** | 0.0000 | 180.70 *** | 0.0000 | 110.90 *** | 0.0000 | 67.12 *** | 0.0000 |
| At most 1 | 158.50 *** | 0.0000 | 100.80 *** | 0.0000 | 100.90 *** | 0.0000 | 63.61 *** | 0.0000 | 57.59 *** | 0.0000 | 37.15 ** | 0.0050 |
| At most 2 | 90.79 *** | 0.0000 | 63.76 *** | 0.0055 | 55.61 *** | 0.0000 | 39.54 *** | 0.0057 | 35.18 *** | 0.0090 | 24.21 | 0.1481 |
| At most 3 | 102.60 *** | 0.0000 | 102.60 *** | 0.0000 | 55.17 *** | 0.0000 | 55.17 *** | 0.0000 | 47.40 *** | 0.0002 | 47.40 *** | 0.0002 |
| Model 2b : $CDE_{it} = f(GDP_{it}, POP_{it}, NREC, \varepsilon_{it})$ | | | | | | | | | | | | |
| None | 382.00 *** | 0.0000 | 272.90 *** | 0.0000 | 281.40 *** | 0.0000 | 209.90 *** | 0.0000 | 100.60 *** | 0.0000 | 63.07 *** | 0.0000 |
| At most 1 | 163.40 *** | 0.0000 | 95.52 *** | 0.0000 | 112.20 *** | 0.0000 | 64.72 *** | 0.0000 | 51.18 *** | 0.0000 | 30.81 ** | 0.0303 |
| At most 2 | 104.20 *** | 0.0000 | 70.72 *** | 0.0010 | 68.48 *** | 0.0000 | 49.85 *** | 0.0002 | 35.74 *** | 0.0076 | 20.87 | 0.2863 |
| At most 3 | 106.10 *** | 0.0000 | 106.10 *** | 0.0000 | 57.86 *** | 0.0000 | 57.86 *** | 0.0000 | 48.25 *** | 0.0001 | 48.25 *** | 0.0001 |
| Model 3a : $CDE_{it} = f(GDP_{it}, POP_{it}, FDINI, \varepsilon_{it})$ | | | | | | | | | | | | |
| None | 342.70 *** | 0.0000 | 258.70 *** | 0.0000 | 238.00 *** | 0.0000 | 181.70 *** | 0.0000 | 158.20 *** | 0.0000 | 118.40 *** | 0.0000 |
| At most 1 | 157.00 *** | 0.0000 | 102.00 *** | 0.0000 | 106.80 *** | 0.0000 | 69.19 *** | 0.0000 | 61.79 *** | 0.0000 | 39.98 *** | 0.0050 |
| At most 2 | 94.71 *** | 0.0000 | 83.32 *** | 0.0000 | 60.37 *** | 0.0000 | 57.02 *** | 0.0000 | 41.36 *** | 0.0033 | 30.52 * | 0.0619 |
| At most 3 | 49.57 * | 0.0990 | 49.57 * | 0.0990 | 22.21 | 0.2229 | 22.21 | 0.2229 | 42.15 *** | 0.0026 | 42.15 *** | 0.0026 |
| Model 3b : $CDE_{it} = f(GDP_{it}, POP_{it}, FDINO, \varepsilon_{it})$ | | | | | | | | | | | | |
| None | 343.7 *** | 0.0000 | 259.9 *** | 0.0000 | 258.4 *** | 0.0000 | 202.3 *** | 0.0000 | 85.36 *** | 0.0000 | 57.61 *** | 0.0000 |
| At most 1 | 147.8 *** | 0.0000 | 105.3 *** | 0.0000 | 101.6 *** | 0.0000 | 78.78 *** | 0.0000 | 46.19 *** | 0.0008 | 26.49 | 0.1502 |
| At most 2 | 85.4 *** | 0.0000 | 71.78 *** | 0.0015 | 47.94 *** | 0.0004 | 42.37 *** | 0.0025 | 37.46 ** | 0.0103 | 29.42 * | 0.0798 |
| At most 3 | 57.16 ** | 0.0384 | 57.16 ** | 0.0384 | 27.53 | 0.1210 | 27.53 | 0.1210 | 29.63 * | 0.0760 | 29.63 * | 0.0760 |

**Table 6.** *Cont.*

| | G20 | | | | Developing | | | | Developed | | | |
|---|---|---|---|---|---|---|---|---|---|---|---|---|
| **Hypothesized** | **Fisher Stat. *** | | **Fisher Stat. *** | | **Fisher Stat. *** | | **Fisher Stat. *** | | **Fisher Stat. *** | | **Fisher Stat. *** | |
| **No. of CE(s)** | **(from Trace Test)** | **Prob.** | **(from Max-Eigen Test)** | **Prob.** | **(from Trace Test)** | **Prob.** | **(from Max-Eigen Test)** | **Prob.** | **(from Trace Test)** | **Prob.** | **(from Max-Eigen Test)** | **Prob.** |
| | | | | | Model 4a : $CDE_{it} = f(GDP_{it}, POP_{it}, SMC, \varepsilon_{it})$ | | | | | | | |
| None | 191.70 *** | 0.0000 | 129.50 *** | 0.0000 | 103.40 *** | 0.0000 | 77.16 *** | 0.0000 | 88.37 *** | 0.0000 | 52.33 *** | 0.0000 |
| At most 1 | 96.40 *** | 0.0000 | 68.23 *** | 0.0001 | 42.37 *** | 0.0000 | 31.59 *** | 0.0016 | 54.03 *** | 0.0000 | 36.65 *** | 0.0058 |
| At most 2 | 57.46 *** | 0.0018 | 44.08 ** | 0.0469 | 23.57 ** | 0.0232 | 18.04 | 0.1146 | 33.88 ** | 0.0130 | 26.05 * | 0.0986 |
| At most 3 | 48.27 ** | 0.0186 | 48.27 ** | 0.0186 | 21.42 ** | 0.0446 | 21.42 ** | 0.0446 | 26.86 * | 0.0817 | 26.86 * | 0.0817 |
| | | | | | Model 4b : $CDE_{it} = f(GDP_{it}, POP_{it}, STV, \varepsilon_{it})$ | | | | | | | |
| None | 361.70 *** | 0.0000 | 272.60 *** | 0.0000 | 257.90 *** | 0.0000 | 202.40 *** | 0.0000 | 103.90 *** | 0.0000 | 70.26 *** | 0.0000 |
| At most 1 | 167.90 *** | 0.0000 | 123.30 *** | 0.0000 | 112.90 *** | 0.0000 | 91.58 *** | 0.0000 | 54.95 *** | 0.0000 | 31.74 ** | 0.0462 |
| At most 2 | 88.01 *** | 0.0000 | 66.72 *** | 0.0027 | 46.06 *** | 0.0003 | 33.47 *** | 0.0146 | 41.94 *** | 0.0028 | 33.25 ** | 0.0317 |
| At most 3 | 69.31 *** | 0.0014 | 69.31 *** | 0.0014 | 37.76 *** | 0.0042 | 37.76 *** | 0.0042 | 31.55 ** | 0.0484 | 31.55 ** | 0.0484 |
| | | | | | Model 5a : $CDE_{it} = f(GDP_{it}, POP_{it}, EXT, \varepsilon_{it})$ | | | | | | | |
| None | 293.60 *** | 0.0000 | 216.20 *** | 0.0000 | 186.50 *** | 0.0000 | 141.70 *** | 0.0000 | 107.10 *** | 0.0000 | 74.51 *** | 0.0000 |
| At most 1 | 135.10 *** | 0.0000 | 81.23 *** | 0.0001 | 78.44 *** | 0.0000 | 46.38 *** | 0.0007 | 56.63 *** | 0.0000 | 34.85 ** | 0.0209 |
| At most 2 | 94.48 *** | 0.0000 | 75.45 *** | 0.0006 | 54.01 *** | 0.0001 | 44.48 *** | 0.0013 | 40.48 *** | 0.0043 | 30.97 * | 0.0555 |
| At most 3 | 64.57 *** | 0.0082 | 64.57 *** | 0.0082 | 34.52 ** | 0.0228 | 34.52 ** | 0.0228 | 30.05 * | 0.0690 | 30.05 * | 0.0690 |
| | | | | | Model 5b : $CDE_{it} = f(GDP_{it}, POP_{it}, IMT, \varepsilon_{it})$ | | | | | | | |
| None | 350.50 *** | 0.0000 | 242.60 *** | 0.0000 | 243.40 *** | 0.0000 | 177.40 *** | 0.0000 | 107.10 *** | 0.0000 | 65.18 *** | 0.0000 |
| At most 1 | 173.90 *** | 0.0000 | 110.60 *** | 0.0000 | 111.70 *** | 0.0000 | 71.02 *** | 0.0000 | 62.23 *** | 0.0000 | 39.62 *** | 0.0056 |
| At most 2 | 109.10 *** | 0.0000 | 85.89 *** | 0.0000 | 66.85 *** | 0.0000 | 52.61 *** | 0.0001 | 42.29 *** | 0.0025 | 33.29 ** | 0.0314 |
| At most 3 | 71.11 *** | 0.0018 | 71.11 *** | 0.0018 | 41.56 *** | 0.0032 | 41.56 *** | 0.0032 | 29.55 * | 0.0775 | 29.55 * | 0.0775 |
| | | | | | Model 6a : $CDE_{it} = f(GDP_{it}, POP_{it}, COE, \varepsilon_{it})$ | | | | | | | |
| None | 296.90 *** | 0.0000 | 226.10 *** | 0.0000 | 186.60 *** | 0.0000 | 153.80 *** | 0.0000 | 110.20 *** | 0.0000 | 72.26 *** | 0.0000 |
| At most 1 | 126.90 *** | 0.0000 | 79.49 *** | 0.0002 | 69.16 *** | 0.0000 | 40.41 *** | 0.0044 | 57.79 *** | 0.0000 | 39.07 *** | 0.0065 |
| At most 2 | 87.39 *** | 0.0000 | 76.91 *** | 0.0004 | 49.63 *** | 0.0003 | 43.88 *** | 0.0016 | 37.76 *** | 0.0095 | 33.03 ** | 0.0335 |
| At most 3 | 53.69 * | 0.0726 | 53.69 * | 0.0726 | 28.95 * | 0.0888 | 28.95 * | 0.0888 | 24.74 | 0.2116 | 24.74 | 0.2116 |
| | | | | | Model 6b : $CDE_{it} = f(GDP_{it}, POP_{it}, COI, \varepsilon_{it})$ | | | | | | | |
| None | 351.00 *** | 0.0000 | 248.00 *** | 0.0000 | 252.90 *** | 0.0000 | 191.00 *** | 0.0000 | 98.11 *** | 0.0000 | 57.06 *** | 0.0000 |
| At most 1 | 179.70 *** | 0.0000 | 122.20 *** | 0.0000 | 121.10 *** | 0.0000 | 89.30 *** | 0.0000 | 58.61 *** | 0.0000 | 32.94 ** | 0.0342 |
| At most 2 | 103.20 *** | 0.0000 | 86.93 *** | 0.0000 | 57.51 *** | 0.0000 | 46.71 *** | 0.0006 | 45.72 *** | 0.0009 | 40.23 *** | 0.0047 |
| At most 3 | 63.49 ** | 0.0105 | 63.49 ** | 0.0105 | 36.30 ** | 0.0142 | 36.30 ** | 0.0142 | 27.19 | 0.1300 | 27.19 | 0.1300 |

Note: The symbols ***, **, and * reveal that the *t*-test is significant at the thresholds of 1%, 5%, and 10%, respectively, which means that the null hypothesis is rejected.

### 4.3. The Long-Run CO_2 Emissions Output Elasticity (FMOLS) Results and Policy Implications

The cointegration test confirmed that there is a long-run equilibrium relationship among these variables. Furthermore, the FMOLS technique in Equations (2) has been explored to generate the long-run elasticities. The empirical results of these models are presented in Table 7.

**Table 7.** FMOLS long-run elasticities.

| G20 | | | Developing | | | Developed | | |
|---|---|---|---|---|---|---|---|---|
| **Variable** | **Coefficient** | **Prob.** | **Variable** | **Coefficient** | **Prob.** | **Variable** | **Coefficient** | **Prob.** |
| | | | | Primary sector | | | | |
| | | | Model 1a : $CDE_{it} = f(GDP_{it}, POP_{it}, AGR, \varepsilon_{it})$ | | | | | |
| GDP | 0.3111 *** | 0.0000 | GDP | 0.2307 *** | 0.0000 | GDP | 0.2147 *** | 0.0000 |
| POP | 0.3838 *** | 0.0000 | POP | 0.5032 *** | 0.0000 | POP | 0.0149 *** | 0.0019 |
| AGR | 0.1992 *** | 0.0000 | AGR | 0.1943 *** | 0.0000 | AGR | −0.0469 *** | 0.0001 |
| R-squared | 0.9782 | | R-squared | 0.9818 | | R-squared | 0.8882 | |
| Adjusted R-squared | 0.9771 | | Adjusted R-squared | 0.9808 | | Adjusted R-squared | 0.8817 | |
| | | | | Secondary sector | | | | |
| | | | Model 1b : $CDE_{it} = f(GDP_{it}, POP_{it}, IND, \varepsilon_{it})$ | | | | | |
| GDP | 0.2134 *** | 0.0000 | GDP | 0.2754 *** | 0.0000 | GDP | 0.3452 *** | 0.0000 |
| POP | 0.2166 *** | 0.0000 | POP | 0.2060 | 0.1731 | POP | 0.1429 *** | 0.0000 |
| IND | 0.4900 *** | 0.0000 | IND | 0.2395 * | 0.0583 | IND | 0.7072 *** | 0.0000 |
| R-squared | 0.9812 | | R-squared | 0.9833 | | R-squared | 0.8960 | |
| Adjusted R-squared | 0.9802 | | Adjusted R-squared | 0.9824 | | Adjusted R-squared | 0.8900 | |
| | | | Model 2a : $CDE_{it} = f(GDP_{it}, POP_{it}, REC, \varepsilon_{it})$ | | | | | |
| GDP | 0.3605 *** | 0.0000 | GDP | 0.2663 *** | 0.0000 | GDP | 0.2877 *** | 0.0000 |
| POP | 0.2342 *** | 0.0000 | POP | 0.2451 *** | 0.0000 | POP | −0.3588 *** | 0.0000 |
| REC | −0.2530 *** | 0.0000 | REC | 0.0430 *** | 0.0000 | REC | −0.1253 *** | 0.0000 |
| R-squared | 0.9762 | | R-squared | 0.9831 | | R-squared | 0.9342 | |
| Adjusted R-squared | 0.9750 | | Adjusted R-squared | 0.9822 | | Adjusted R-squared | 0.9307 | |
| | | | Model 2b : $CDE_{it} = f(GDP_{it}, POP_{it}, NREC, \varepsilon_{it})$ | | | | | |
| GDP | 0.0643 *** | 0.0000 | GDP | 0.0731 *** | 0.0000 | GDP | 0.1273 *** | 0.0000 |
| POP | −1.2809 *** | 0.0000 | POP | −1.1344 *** | 0.0000 | POP | −1.1071 *** | 0.0000 |
| NREC | 0.9074 *** | 0.0000 | NREC | 0.7970 *** | 0.0000 | NREC | 1.1017 *** | 0.0000 |
| R-squared | 0.9956 | | R-squared | 0.9935 | | R-squared | 0.9841 | |
| Adjusted R-squared | 0.9954 | | Adjusted R-squared | 0.9932 | | Adjusted R-squared | 0.9832 | |
| | | | | Tertiary sector | | | | |
| | | | Model 3a : $CDE_{it} = f(GDP_{it}, POP_{it}, FDINI, \varepsilon_{it})$ | | | | | |
| GDP | 0.2346 *** | 0.0000 | GDP | 0.1752 *** | 0.0000 | GDP | 0.1257 *** | 0.0000 |
| POP | 0.2171 *** | 0.0000 | POP | 0.4131 *** | 0.0000 | POP | −0.0477 *** | 0.0000 |
| FDINI | 0.0269 *** | 0.0000 | FDINI | −0.0231 *** | 0.0011 | FDINI | 0.0054 | 0.7919 |
| R-squared | 0.9760 | | R-squared | 0.9783 | | R-squared | 0.9283 | |
| Adjusted R-squared | 0.9747 | | Adjusted R-squared | 0.9770 | | Adjusted R-squared | 0.9243 | |

| G20 | | | Developing | | | Developed | | |
|---|---|---|---|---|---|---|---|---|
| **Variable** | **Coefficient** | **Prob.** | **Variable** | **Coefficient** | **Prob.** | **Variable** | **Coefficient** | **Prob.** |
| | | | Model 3b : $CDE_{it} = f(GDP_{it}, POP_{it}, FDINO, \varepsilon_{it})$ | | | | | |
| GDP | 0.1990 *** | 0.0000 | GDP | 0.1443 *** | 0.0000 | GDP | 0.0755 *** | 0.0000 |
| POP | 0.1487 *** | 0.0000 | POP | 0.3314 *** | 0.0000 | POP | 0.0316 *** | 0.0000 |
| FDINO | −0.0096 | 0.2954 | FDINO | 0.0041 | 0.7447 | FDINO | −0.0950 *** | 0.0005 |
| R-squared | 0.9763 | | R-squared | 0.9748 | | R-squared | 0.8956 | |
| Adjusted R-squared | 0.9750 | | Adjusted R-squared | 0.9732 | | Adjusted R-squared | 0.8898 | |
| | | | Model 4a : $CDE_{it} = f(GDP_{it}, POP_{it}, SMC, \varepsilon_{it})$ | | | | | |
| GDP | 0.0306 ** | 0.0135 | GDP | 0.2560 *** | 0.0000 | GDP | 0.1648 *** | 0.0000 |
| POP | −0.0489 *** | 0.0000 | POP | 0.0030 | 0.0000 | POP | −0.0386 *** | 0.0000 |
| SMC | 0.0384 *** | 0.0000 | SMC | 0.0005 | 0.0000 | SMC | −0.0271 ** | 0.0711 |
| R-squared | 0.9810 | | R-squared | 0.9883 | | R-squared | 0.9420 | |
| Adjusted R-squared | 0.9798 | | Adjusted R-squared | 0.9874 | | Adjusted R-squared | 0.9385 | |
| | | | Model 4b : $CDE_{it} = f(GDP_{it}, POP_{it}, STV, \varepsilon_{it})$ | | | | | |
| GDP | 0.2464 *** | 0.0000 | GDP | 0.1427 *** | 0.0000 | GDP | 0.1956 *** | 0.0000 |
| POP | 0.0663 *** | 0.0000 | POP | 0.1741 *** | 0.0000 | POP | −0.1025 *** | 0.0000 |
| STV | −0.0164 | 0.5718 | STV | 0.1089 *** | 0.0000 | STV | 0.0232 *** | 0.0008 |
| R-squared | 0.9822 | | R-squared | 0.9689 | | R-squared | 0.9338 | |
| Adjusted R-squared | 0.9812 | | Adjusted R-squared | 0.9669 | | Adjusted R-squared | 0.9300 | |
| | | | Model 5a : $CDE_{it} = f(GDP_{it}, POP_{it}, EXT, \varepsilon_{it})$ | | | | | |
| GDP | 0.2358 *** | 0.0000 | GDP | 0.2187 *** | 0.0000 | GDP | 0.1174 *** | 0.0000 |
| POP | 0.0740 *** | 0.0000 | POP | −0.0035 * | 0.0504 | POP | 0.0409 *** | 0.0000 |
| EXT | 0.0817 *** | 0.0000 | EXT | 0.1973 *** | 0.0000 | EXT | −0.0059 | 0.7405 |
| R-squared | 0.9786 | | R-squared | 0.983972 | | R-squared | 0.9277 | |
| Adjusted R-squared | 0.9776 | | Adjusted R-squared | 0.98311 | | Adjusted R-squared | 0.9238 | |
| | | | Model 5b : $CDE_{it} = f(GDP_{it}, POP_{it}, IMT, \varepsilon_{it})$ | | | | | |
| GDP | 0.2495 *** | 0.0000 | GDP | 0.1551 *** | 0.0000 | GDP | 0.1271 *** | 0.0000 |
| POP | 0.1306 *** | 0.0000 | POP | 0.1090 *** | 0.0000 | POP | 0.0241 *** | 0.0000 |
| IMT | 0.0337 *** | 0.0000 | IMT | 0.2508 *** | 0.0000 | IMT | 0.0487 *** | 0.0000 |
| R-squared | 0.9781 | | R-squared | 0.9813 | | R-squared | 0.9222 | |
| Adjusted R-squared | 0.9770 | | Adjusted R-squared | 0.9803 | | Adjusted R-squared | 0.9179 | |
| | | | Model 6a : $CDE_{it} = f(GDP_{it}, POP_{it}, COE, \varepsilon_{it})$ | | | | | |
| GDP | 0.2430 *** | 0.0000 | GDP | 0.2583 *** | 0.0000 | GDP | 0.1814 *** | 0.0000 |
| POP | 0.2293 *** | 0.0000 | POP | 0.1985 *** | 0.0000 | POP | −0.0110 ** | 0.0164 |
| COE | −0.0457 *** | 0.0000 | COE | 0.0636 *** | 0.0000 | COE | −0.0999 *** | 0.0000 |
| R-squared | 0.9782 | | R-squared | 0.9839 | | R-squared | 0.9253 | |
| Adjusted R-squared | 0.9772 | | Adjusted R-squared | 0.9830 | | Adjusted R-squared | 0.9213 | |
| | | | Model 6b : $CDE_{it} = f(GDP_{it}, POP_{it}, COI, \varepsilon_{it})$ | | | | | |
| GDP | 0.2213 *** | 0.0000 | GDP | 0.2613 *** | 0.0000 | GDP | 0.1349 *** | 0.0000 |
| POP | 0.2056 *** | 0.0000 | POP | 0.2002 *** | 0.0000 | POP | 0.0237 *** | 0.0000 |
| COI | 0.0617 *** | 0.0000 | COI | 0.1808 *** | 0.0000 | COI | −0.0152 **** | 0.0645 |
| R-squared | 0.9780 | | R-squared | 0.9839 | | R-squared | 0.9255 | |
| Adjusted R-squared | 0.9769 | | Adjusted R-squared | 0.9831 | | Adjusted R-squared | 0.9215 | |

Note: The symbols *** and ** reveal that the t-test is significant at the thresholds of 1%, 5%, and 10%, respectively.

The results in Models 1a and 1b imply that for every 1% increase in industry, $CO_2$ emissions will increase by 0.4900%, 0.2395% and 0.7072% for the full sample of G20 countries and for the subsamples of developing and developed economies, respectively. Here, the developed economies show a 3 times higher increase than the developing economies in the G20. On the other hand, for every 1% increase in agriculture, $CO_2$ emissions decrease by 0.0469% in developed economies, while they increase by 0.1943% in developing economies. This result implies that $CO_2$ emissions are more sensitive in the agricultural sector of developed countries than in that of developing economies. Moreover, the role of agriculture in $CO_2$ emissions is not consistent across developed and developing economies.

The output elasticities in Models 2a and 2b show that for every percentage point of energy use that falls into the category of nonrenewable energy, $CO_2$ emissions will increase by 0.9074%, 0.7979%, and 1.1017% for the full sample and the subsamples of developing and developed economies, respectively. A 1% rise in renewable energy consumption reduces $CO_2$ emissions by 0.2530% and 0.1253% in the full sample and among the subsample of developed economies, respectively, while it raises $CO_2$ emissions by 0.0430% for developing economies in the G20. The result implies that nonrenewable energy consumption is significantly positively related to $CO_2$ emissions across the panel. Moreover, renewable energy has a positive effect on $CO_2$ emissions in developing countries.

The results in Models 3a and 3b imply that a 1% increase in FDI inflows decreases the $CO_2$ emissions of developing economies by 0.0231%; on the other hand, a 1% increase in FDI outflows decreases the $CO_2$ emissions of developed economies by 0.095%. The results show that developing and developed economies experience different degrees of $CO_2$ reduction.

The output elasticities, as per Models 4a and 4b, show that for every 1% increase in the stock market capitalization of listed domestic companies (SMC), $CO_2$ emissions will decrease by 0.0271% in developed economies within the G20, while they will increase by 0.0005% in developing economies within the G20. Stock trading volume (STV) is positively related to $CO_2$ emissions in both developing and developed economies, increasing emissions by 0.1089% and 0.0232%, respectively. This finding implies that developing stronger capital markets will help reduce $CO_2$ emissions, whereas stock trading numbers have a significantly positive relationship with $CO_2$ emissions across the panel.

The results in Models 5a and 5b show that a 1% increase in exports raises $CO_2$ emissions by 0.0817% and 0.1973% in the full G20 sample and the subsample of developing economies, respectively. A 1% increases in imports increases $CO_2$ emissions in the full sample and subsamples of developing and developed economies by 0.0337%, 0.2508% and 0.0478%, respectively. The elasticity gap of imports with $CO_2$ emissions is very large between developing and developed economies, and in developing economies, it is 5 times more than that in developed economies. This result implies that exports will be accompanied by $CO_2$ emissions during processing in all G20 economies, especially in developing economies. Moreover, within the G20 countries, developed economies import more environmentally friendly products than developing economies.

The results in Models 6a and 6b imply that a 1% increase in commercial service exports (COE) will decrease $CO_2$ emissions by 0.1% and 0.047% in developed countries and in the full G20 sample, respectively, while increasing them by 0.0636% in developing economies. Similarly, a 1% increase in commercial service imports (COI) will increase $CO_2$ emissions by 0.1808% for developing economies and reduce them by 0.0152% for developed economies. This result implies that the role of both commercial service exports and commercial service imports on $CO_2$ emissions is consistent across the developed and developing economies of the G20.

### 4.3.1. Primary Sector and $CO_2$ Emissions

Agriculture in the primary sector in developed economies is working in favor of reducing $CO_2$ emissions; however, agriculture has a positive impact on $CO_2$ emissions in the case of developing economies. This simultaneously validates the EKC hypothesis, which can

be interpreted to mean that the modernization of farming practices will not only increase economic growth by improvements to inter-sector and intra-sector productivity but also optimize energy efficiency and increase energy productivity, causing a reduction in $CO_2$ emissions. Therefore, we suggest that policymakers in developing economies upgrade productive infrastructure, increase land consolidation and modernization, accumulate capital and increase investment in science, research and technology capacity in the primary sector, while developed countries need to extend their experience and advance their technology in productivity and technology efficiency globally.

Our findings also prove that the development of agriculture is in line with the EKC hypothesis that a more developed economy will improve environmental degradation. Technology, productivity and energy efficiency will be upgraded alongside rapid economic development, thus reducing energy consumption to stimulate economic growth and making the primary sector more environmentally friendly.

### 4.3.2. Secondary Sector and $CO_2$ Emissions

The impact of the secondary sector on $CO_2$ emissions can be divided into two groups. The first group consists of variables related to industry that are significantly positively related to $CO_2$ emissions for developing and developed economies in the G20, such as industry and nonrenewable consumption. Since developed economies show a 3 times greater increase in $CO_2$ emissions than developing economies in the G20 for every 1% increase in industry, we can confirm that the industry sector in developed countries is more sensitive to $CO_2$ emissions than that in developing countries. This finding implies that developed economies assemble and manufacture many products in developing economies, while they are responsible only for designing and supervising the production of these products. As a result, a small increase in industry would have a strong impact on $CO_2$ emissions. This finding can also be interpreted to mean that the secondary sector supports both the primary sector and the tertiary sector with light industry and heavy industry. Therefore, we suggest that policymakers in developed and developing economies increase productivity, resulting in higher labor productivity and energy efficiency and leading to more environmentally friendly industries.

The second group of variables in developing economies serves to aggravate $CO_2$ emissions; however, it has a negative effect in the case of developed economies. Renewable energy consumption has an effect on $CO_2$ emissions in developing economies completely opposite to that in developed economies. We can confirm that renewable energy has a positive effect on $CO_2$ emissions in developing countries. This phenomenon validates the EKC hypothesis and can be explained by the fact that solid biofuels for traditional use are the largest renewable sources for most of the developing economies in the G20; however, almost 0% of developed economies in the G20 use this traditional source of renewable energy. This finding implies that developing economies should change their energy structure by replacing more non-clean renewable energy with clean renewable energy, while policymakers in developed economies need to consider $CO_2$ emissions from a global perspective by extending advanced technology pertaining to renewable energy development to all countries.

This finding also demonstrates that as the economy grows, energy efficiency and clean energy technologies will improve, thus improving the environment and making secondary industry more environmentally friendly.

### 4.3.3. Tertiary Sector and $CO_2$ Emissions

The effects of the tertiary sector on $CO_2$ emissions can be divided into three groups. The first group comprises the variables that favor increasing $CO_2$ emissions in developing economies and decreasing $CO_2$ emissions in developed economies. Stock market capitalization (SMC) represents a government's policy orientation and market demand. Economic growth is supported mainly by higher productivity and higher energy efficiency in developed markets; thus, this demand will be reflected in the stock market. Moreover, developed

economies are more efficient and have strong stock markets in which the price of new technology and higher productivity will be directly reflected in share prices. Thus, the stock market itself is an energy efficiency filter; in contrast, developing economies focus more on the expansion of economic growth, not on $CO_2$ reduction. On the one hand, compared with that in a developed market, the price of new technology and higher productivity in a developing market needs time to ferment and does not have the ability to directly reflect share prices. Policymakers in developing economies need to increase the efficiency of their stock markets. Furthermore, a healthy financial system, including a healthy stock market, can help firms resolve funding issues in both the short term and the long term.

Both commercial service exports and imports show a positive relationship with $CO_2$ emissions in developing economies and a negative relationship in developed economies. This simultaneously validates the EKC hypothesis; as mentioned above, the commercial service gap between developing and developed economies was more than USD 1000 billion in 2018. This implies that information and communication technologies not only play an important role in economic growth but also help increase productivity in manufacturing industries, resulting in a reduction in $CO_2$ emissions. Policymakers in developing economies need to improve information and communications technology, facilitating infrastructure and fulfilling the need for training of skilled labor.

Additionally, exports show a positive relationship with $CO_2$ emissions in developing economies. This is because developing economies need more raw materials for manufacturing processes, and developed economies manufacture their products in developing economies. This finding implies that developing economies import $CO_2$ in exchange for economic growth.

The second group consists of variables related to stock trading value and imports, which are significantly positively related to $CO_2$ emissions in both developing and developed economies. Stock trading value is the total number of domestic and foreign shares traded, multiplied by their respective matching prices. The trading of stocks does not directly generate an increase in social wealth but is a transfer or reallocation of wealth among shareholders. Without a stock market, the value of an enterprise would be severely affected by its illiquidity. Thus, because a high trading value in stock would increase $CO_2$ emissions, we suggest that policymakers in both economies consider applying an environmental protection tax in proportion to the trading value.

The third group contains variables that work in favor of reducing $CO_2$ emissions in only one type of economy. FDI inflows have a negative impact on $CO_2$ emissions in developing economies, and FDI outflows have a negative impact on $CO_2$ emissions in developed economies. This finding can be interpreted to mean that FDI inflows bring new technology into host countries or direct imports of new technology into a country where the technology may not be available. However, this is not the case in developed economies, as they do not rely on FDI inflows for technology transfer and innovation. Moreover, these foreign investors might not pay much attention to emissions control during production. Therefore, we suggest that policymakers in developing economies attract more FDI inflows from developed economies by reducing taxes and paying benefits to GHG emission reduction projects. On the other hand, we suggest that policymakers in developed economies promote policies that convert FDI into clean and renewable energy projects.

As listed in Table 8, nearly all variables in the tertiary sector are positively related to $CO_2$ emissions in developing countries and negatively related to $CO_2$ emissions in developed countries. This simultaneously validates the EKC hypothesis. The tertiary sector in developed countries is fully resistant to environmental degradation by virtue of more advanced technology and better energy use efficiency.

**Table 8.** FMOLS long-run results for the three-sector model.

| Variables | Developing Economies | Developed Economies |
|---|---|---|
| | Primary sector | |
| AGR | + | − |
| | Secondary sector | |
| IND | + | + |
| REC | + | − |
| NREC | + | + |
| | Tertiary sector | |
| FDINI | - | null |
| FDINO | null | − |
| SMC | + | − |
| STV | + | + |
| EXP | + | null |
| IMP | + | + |
| COE | + | − |
| COI | + | − |

These findings and policy suggestions address the research gap and illustrate that, as opposed to the many controversial validation formulas for determining the validity of EKC, the three-sector model of validation allows the effects of EKC to be combined with real industries to identify their patterns.

**5. Conclusions**

This study examined the validity of the EKC hypothesis by verifying the relevance of the three-sector model for developing and developed countries. The empirical findings of this study suggest that there is a significant long-run equilibrium relationship between the three sectors through 12 variables and $CO_2$ emissions. This finding implies that $CO_2$ emissions are closely related to primary, secondary, and tertiary sectors in the long run. The long-run $CO_2$ emissions elasticities indicate that the primary sector (agriculture) has a positive and negative effect on the $CO_2$ emissions in developing and developed economies, respectively. In the secondary sector, three variables show a positive relationship with the $CO_2$ emissions of developing economies: industry, nonrenewable energy consumption and renewable energy consumption; while for developed economies, there are two variables: industry and nonrenewable energy consumption. Except for FDI inflow, all variables in the tertiary sector for developing economies show a positive relationship with $CO_2$ emissions, including commercial service exports, commercial service imports, stock market capitalization, stock trading value, exports and imports. All variables in the tertiary sector of developed economies show a negative relationship with $CO_2$ emissions except for stock trading value and imports.

Furthermore, the EKC hypothesis has been confirmed. The relationship between the three sectors in the economy and carbon dioxide emissions shows opposite results for the developing and developed G20 countries. Developed countries, with their more advanced economies, production technologies, and clean energy technologies, better energy efficiency and more competitive stock markets, are well positioned to improve the environmental degradation caused by $CO_2$ emissions. This finding is direct evidence that economic development can improve the environment. Future research could be concentrated in the following areas: (1) expansion of the study beyond the G20 to more countries, regions and economic groupings; (2) extension of the research time interval; (3) the selection of

additional environmental indicators as dependent variables, i.e., $NO_2$, $SO_2$, PM diameter; and (4) the addition of more explanatory variables.

**Author Contributions:** Conceptualization, K.Y., R.G. and V.W.; methodology, K.Y.; software, K.Y.; validation, K.Y.; formal analysis, K.Y.; investigation, K.Y.; resources, K.Y.; data curation, K.Y.; writing-original draft preparation, K.Y.; writing-review and editing, K.Y., R.G. and V.W.; visualization, K.Y.; supervision, R.G. and V.W.; project administration, R.G. and V.W. All authors have read and agreed to the published version of the manuscript.

**Funding:** This research received no external funding.

**Institutional Review Board Statement:** Not applicable.

**Informed Consent Statement:** Not applicable.

**Data Availability Statement:** Publicly archived World Bank database: https://data.worldbank.org/indicator/EN.ATM.CO2E.PC?view=chart.

**Acknowledgments:** We would like to express our sincere gratitude to the anonymous reviewers, whose suggestions were very professional and constructive; this paper has been made more complete and better because of their suggestions. They have given us a lot of support and help. Such support is what the article and the authors need most. We are very grateful.

**Conflicts of Interest:** The authors declare no conflict of interest.

## Abbreviations

| | |
|---|---|
| $CO_2$ | (carbon dioxide) |
| EKC | (Environment Kuznets Curve) |
| FDI | (foreign direct investment) |
| CDE | ($CO_2$ emissions) |
| NREC | (nonrenewable energy consumption) |
| REC | (renewable energy consumption) |
| EXT | (export of goods and services) |
| IMT | (import of goods and services) |
| COE | (commercial service exports) |
| COI | (commercial service imports) |
| FDI | (foreign direct investment) |
| FDINI | (foreign direct investment net inflows) |
| FDINO | (foreign direct investment net outflows) |
| SMC | (stock market capitalization) |
| STV | (stock trade total value) |
| AGR | (Agriculture) |
| IND | (industry) |
| STIRPAT | (Stochastic Impacts by Regression on Population, Affluence and Technology) |
| CADF | (cross-section augmented Dickey-Fuller) |
| IPS | (Im et al. (2003) panel unit root test) |
| FMOLS | (Fully Modified ordinary least squares) |

## Notes

[1] $ln(E/P)_{it} = \alpha_t + \beta_{1t}ln(GDP/P)_{it} + \beta_{2t}(ln(GDP/P))^2_{it} + \varepsilon_{it}$. where $E$ is the environmental indicator, i.e., $CO_2$ emissions (Shuai et al. 2017), water chemical oxygen and $SO_2$ emissions (Jayanthakumaran et al. 2012), biodiversity conservation and biological capacity (Mills and Waite 2009). $P$ is the population of the specific study region. GDP is the gross domestic product of a specific study region and is used to indicate the level of income.

[2] Primary sector: agriculture; Secondary sector: industry, renewable energy, nonrenewable energy; Tertiary sector: export, import, FDI inflow, FDI outflow, commercial service exports, and commercial service imports, stock market capitalization, stock market trading value.

[3] The data for 2014 will be used until this study is finalized.

[4] According to the World Bank (2021), commercial services are defined as "the intangible product that is produced, transferred and consumed accompanied with economic output".

5    (1) Data are from the World Bank database. (2) Data are missing for India for 2018.
6    (1) Data are from the World Bank database.
7    Data from 2014 will be used until this study is finalized; World Bank database. https://data.worldbank.org/indicator/EN.ATM.CO2E.PC?view=chart (accessed on 10 June 2021).
8    None of the data used are percentages.
9    Nonrenewable energy consumption (NREC); renewable energy consumption (REC); export of goods and services (EXT); import of goods and services (IMT); commercial service exports (COE); commercial service imports (COI); foreign direct investment net inflows (FDINI); foreign direct investment net outflows (FDINO); agriculture (AGR); industry (IND); stock market capitalization (SMC); stock trade total value (STV).

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
