# Peer review of "CO2 Emissions in G20 Nations through the Three-Sector Model"

_jrfm, doi:10.3390/jrfm15090394_

Round 1

Reviewer 1 Report

Comment 1: Abstract should inform about the adopted methodology and how objective is achieved rather than merely discussing the insight of the results.

Comment 2: It will be interesting to see novelty and knowledge addition in a separate paragraph.

Comment 3: The knowledge gap needs to be clearly addressed in a separate paragraph.

Comment 4: One can also observe that meticulous literature study is missing. Therefore it is recommended to include some more research paper in particular to the scope of study.

Comment 5: Author(s) should add the future scope and enhancement at the end.

Comment 6: All the Equations, tables and figures should be self-evident.

Comment 7: Please check the manuscript from top to bottom for typing and grammatical errors.

Comment 8:  Abbreviations should be defined in the text when first time used and their use should be minimized, if not required.

Comment 9: The Methodology should be concise and logical allowing interested researchers to be able to repeat your work. If the methodology, or some parts, has been already published elsewhere, you should summarize it and provide reference.

Comment 10: Please clarify the meaning of each symbol in the formula.

Author Response

Response:

Dear Examiner,

Thank you for your comments on the work. These comments are very important for the quality of this paper. I have revised the paper based on the comments and I have proofread my response to each comment in the following section, with explanations and changes made based on the comments.

Reviewer 2 Report

This paper is studying and testing the hypothesis of the Environment Kuznets Curve.

Table 1 shows simple means in percentage. Each country's population, GDP, and CO2 emission volume are different. Does this make sense?

Tables 2 and 3 have a large variability of values.

US, Russia, China, and India have huge CO2 emissions. Is the reason population, GDP, or regulation policy? 

Did this paper analyze the supply chain?

Is GDP equitable with energy consumption?

How about the energy mix of each country?

Author Response

(The authors gave the same response as above.)
